# Specific NLRP3 Inflammasome Assembling and Regulation in Neutrophils: Relevance in Inflammatory and Infectious Diseases

**DOI:** 10.3390/cells11071188

**Published:** 2022-04-01

**Authors:** Christophe Paget, Emilie Doz-Deblauwe, Nathalie Winter, Benoit Briard

**Affiliations:** 1Centre d’Etude des Pathologies Respiratoires, UMR1100, Inserm (Institut National de la Santé et de la Recherche Médicale), 37012 Tours, France; 2Faculté de Médecine, Université de Tours, 37012 Tours, France; 3INRAe (Institut National de la Recherche pour l’Agriculture, l’Alimentation et l’Environnement), Université de Tours, ISP, 37380 Nouzilly, France; emilie.doz@inrae.fr (E.D.-D.); nathalie.winter@inrae.fr (N.W.)

**Keywords:** inflammasome, neutrophils, NLRP3, caspase-1, caspase-11, IL-1β, IL-18, pyroptosis, gasdermin, Gasdermin-D, NETosis, neutrophil extracelluar traps (NETs), apoptosis, cell death, macrophage, monocyte, cryopyrin-associated periodic syndrome (CAPS)

## Abstract

The NLRP3 inflammasome is a cytosolic multimeric protein platform that leads to the activation of the protease zymogen, caspase-1 (CASP1). Inflammasome activation mediates the proteolytic activation of pro-inflammatory cytokines (IL-1β and IL-18) and program cell death called pyroptosis. The pyroptosis is mediated by the protein executioner Gasdermin D (GSDMD), which forms pores at the plasma membrane to facilitate IL-1β/IL-18 secretion and causes pyroptosis. The NLRP3 inflammasome is activated in response to a large number of pathogenic and sterile insults. However, an uncontrolled inflammasome activation may drive inflammation-associated diseases. Initially, inflammasome-competent cells were believed to be limited to macrophages, dendritic cells (DC), and monocytes. However, emerging evidence indicates that neutrophils can assemble inflammasomes in response to various stimuli with functional relevance. Interestingly, the regulation of inflammasome in neutrophils appears to be unconventional. This review provides a broad overview of the role and regulation of inflammasomes—and more specifically NLRP3—in neutrophils.

## 1. The NLRP3 Inflammasome

The innate immune system is a host defense mechanism conserved through the evolutionary process [1]. Innate immunity relies on a complex network of specialized cells, including myeloid, lymphoid lineage, but also non-hematopoietic cells, to respond to harmful stimuli, such as infection or imbalanced cellular function [1]. This recognition sets into motion multiple innate mechanisms, including phagocytosis, cytotoxicity, programmed cell death or secretion of diverse factors: cytokines, chemokines, and alarmins. Innate immune cells bear receptors to sense pathogen-associated molecular patterns (PAMPs) and damage-associated molecular patterns (DAMPs) produced by the infected or damaged tissues. Engagement of these receptors enables the host to engage an appropriate molecular and cellular response to control the infection. This recognition occurs through a set of germline-encoded pattern recognition receptors (PRRs) expressed at the plasma membrane and cytosol. The membrane-bound receptors are mainly composed of Toll-like receptors (TLRs) and C-type lectin receptors (CLRs). The cytosolic PRRs include the nucleotide-binding oligomerization domain (NOD)-like receptors (NLRs), absent in melanoma 2 (AIM2)-like receptors (ALRs) and retinoic acid-inducible gene I (RIG-I)-like receptors (RLRs) [2,3,4,5]. These cytosolic receptors primarily mediate the production of type I interferon or mediate inflammasome activation [2,4,6].

Inflammasomes are one of the major arms of the innate immune system with crucial roles in microbial infections, cancer, and inflammatory disorders [3,5,7,8,9]. They are multimeric protein platforms that lead to inflammatory protease zymogen caspase-1 activation, which drives proteolytic maturation of pro-inflammatory cytokines IL-1β and IL-18 (Figure 1). In parallel, active caspase-1 induces the cleavage and activation of Gasdermin D (GSDMD). The N-terminal pyroptosis-inducing domain, GSDMD^NT^, can oligomerize and form pores in the plasma membrane, leading to a lytic form of cell death called pyroptosis, which may also facilitate the extracellular release of active IL-1β and IL-18 (Figure 1) [10,11]. Pyroptotic cell death is crucial for the innate immune response to control infection, tumor progression, or inflammatory disorders. To date, multiple innate sensors with the potential to assemble inflammasome complexes have been identified, including the NLR family pyrin domain-containing 1b (NLRP1b), NLR family CARD domain-containing protein 4 (NLRC4), and NLR family pyrin domain-containing 3 (NLRP3), absent in melanoma-2 (AIM2) receptor and pyrin receptor (PYRIN) but also the most recent NLR family pyrin domain-containing 6 (NLRP6) and NLR family pyrin domain-containing 9 (NLRP9) (Figure 1A) [5]. These receptors are capable of sensing PAMPs and DAMPs to assemble inflammasome complexes. Upon sensing and activation, these receptors nucleate the formation of the inflammasome complex by actively recruiting the adaptative molecule called apoptosis-associated speck-like protein-containing CARD (ASC) via domain-domain (PYRIN-PYRIN) interaction (Figure 1A). However in rare cases, as with NLRC4 or NLRP1b, inflammasome may function independently of ASC [12,13,14]. Next, the ASC from these complex recruits and interacts with the caspase-1 CARD domain, resulting in the assembly of a functionally mature inflammasome. Among the well-established inflammasomes, NLRP1 senses the toxin *Bacillus anthracis* anthrax and the secreted protein IpaH7.8 from *Shigella flexneri*, whereas NAIP-NLRC4 responds to bacterial flagellin and the type III secretion system (T3SS) [15,16,17,18]. AIM2 is a DNA sensor that can engage inflammasome formation upon binding to microbial DNA [2,3,5]. PYRIN inflammasome is activated in response to the bacterial Rho-inactivating toxins like *Clostridium difficile* [19,20].

Of all, NLRP3 inflammasome is the most documented as a broad sensor of cellular damage, which responds to several stimuli from various pathogens or resultsfrom metabolic disorders including cancer. 

The NLRP3 inflammasome is activated by two principal mechanisms named canonical and non-canonical activation (Figure 1B,C). The NLRP3 inflammasome has been actively reviewed; the readers should refer to extensive reviews for more comprehensive details concerning the regulation of NLRP3 inflammasome activation [5,9,21,22,23,24].

The canonical NLRP3 inflammasome activation needs two signals: (1) First, a priming signal to promote the expression of the inflammasome components such as NLRP3 as well as the immature inflammatory cytokines pro-IL-1β or pro-IL-18 and (2) a subsequent activation signal such as extracellular ATP or bacterial toxins to promote the oligomerization and activation of NLRP3 (Figure 1B). Despite NLRP3 inflammasome being thoroughly investigated, the complete molecular mechanisms driving NLRP3 activation have not been deciphered yet. Multiple mechanisms are proposed to trigger NLRP3 inflammasome assembly, such as aberrant cellular ionic exchange, lysosomal rupture, mitochondria destabilization, trans-Golgi network disassembly, or translation inhibition, but these mechanisms may differ according to the triggering signal and cell type (Figure 1B). Despite 15 years of intense investigation, the canonical NLRP3 inflammasome is not fully understood [5,25].

The non-canonical NLRP3 inflammasome activation occurs upon Gram-negative bacteria infection or circulating lipopolysaccharide (LPS) (Figure 1C). In this context, caspase-11 (in mice) or caspase-4 and caspase-5 (in humans) mediate NLRP3 activation and caspase-1 cleavage upon LPS sensing (Figure 1C). The release of LPS into the cytoplasm binds to caspase-11 or caspase-4/-5 to initiate their oligomerization (Figure 1C) [26]. Mechanistically, LPS binding to caspase-11 induces oligomerization and proteolytic cleavage of the GSDMD protein [26].

For a while, inflammasomes were considered to be functionally relevant in a restricted set of myeloid cells, including macrophages, monocytes, and dendritic cells. Conversely, in neutrophils, which are key immune sentinels, activation of inflammasomes was considered important [27]. Being traditionally considered as a simple antimicrobial population, the vision of neutrophils has largely evolved over the past two decades. It is now clear that neutrophils are more complex and can mediate a wide range of specialized functions, positioning them as critical cellular components in immunoregulation. Among those emerging functions, neutrophils were identified to engage an unconventional inflammasome with the formation of the supramolecular complex, and release of pro-inflammatory cytokines (IL-1β and IL-18) in absence of classical pyroptosis [27]. This new function for neutrophils is further discussed below.

## 2. Neutrophils: From Phagocytosis Sentinel to Immunoregulation Commander

Neutrophils are the most abundant circulating cell population of the immune system. In humans, neutrophils represent up to 70% of circulating leukocytes and are continuously generated in the bone marrow [28]. It is estimated that 10^11^ neutrophils are daily released into the bloodstream, a figure that drastically increases during infection or inflammation. Of note, a large proportion of “marginated” neutrophils populates peripheral organs. This likely enables a rapid mobilization of mature neutrophils in the case of harmful invaders [29]. Neutrophils derive from a common bone marrow committed myeloid progenitor cell, called hematopoietic stem cell (HSC) (granulocyte-monocyte progenitors), and undergo differentiation and maturation upon stimulation with the cardinal growth factor granulocyte colony-stimulating factor (G-CSF).

Initially, neutrophils were considered as short-lived cells with a circulating half-life of 8 h [30,31,32]. However, the lifespan of neutrophils can significantly increase upon activation [33,34]. Indeed, neutrophils can undergo an anti-apoptotic program to survive longer during inflammation and/or infection [35]. In addition, mature neutrophils can proliferate in the periphery following infection [36]. Furthermore, it has been revealed that extravasated neutrophils could re-enter circulation and, thus, disseminate inflammation to distant sites. Neutrophils are defined based on their polylobed nucleus, various granules, and a set of surface markers. Typically, CD11b, CD66b, CD16, and CD15 expression enables identification of human neutrophils, while co-expression of CD11b and Ly6G are used to identify mouse neutrophils.

Historically, neutrophils were believed to represent a homogeneous population that was fully equipped to exert various potent antimicrobial effector functions to eliminate pathogens and prevent their dissemination. Among those functions, neutrophils are robust phagocytes that can encapsulate microbes in phagosomes [37]. Once phagocyted, neutrophils can exert antimicrobial activities through both oxygen-dependent and -independent pathways. First, they produce several oxidants including the reactive oxygen species (ROS) via a network of enzymes including the NADPH oxidase complex and the myeloperoxidase (MPO). Lack or dysfunction of these enzymes results in chronic granulomatous disease, with a significant predisposition to bacterial and fungal infection [38,39,40]. In parallel, neutrophils can release a myriad of proteases stored in their granules such as cathepsins, defensins, lactoferrin, and lysozyme [41]. This arsenal is then released either into the phagosomes or the extracellular environment, thus acting on both intra- or extracellular pathogens. In addition, neutrophils can eliminate extracellular microorganisms by releasing neutrophil extracellular traps (NETs) in the environment, which acts as a fishnet to trap invaders. The NET structure is mainly composed of DNA and histones but also covered with neutrophil proteases and many other bactericidal proteins and granules [42]. Interestingly, the neutrophil chromatin undergoes a fast decondensation associated with histone modification. Most NET DNA originates from the nucleus, but some mitochondrial DNA can be isolated [42].

Since NET discovery, the list of pathologies reporting NET formation is expanding: NETs are involved in chronic inflammatory diseases such as atherosclerosis [43], autoimmune diseases such as rheumatoid arthritis [44], systemic lupus erythematosus [45], and asthma [46]. Although NETs protect against pathogens, they can also generate tissue damages during infection and inflammatory diseases as recently exemplified in severe COVID-19 patients [47]. They also play a detrimental role in bacterial sepsis [48] and are highly prominent in the sputum of patients with cystic fibrosis [49]. During tuberculosis, they are involved in severe lung damages in susceptible mouse models [50] and patients suffering from active disease [51]. NET release occurs via a programmed cell death (PCD) mechanism termed NETosis [52]. Interestingly, non-lytic NETosis is an alternative mechanism in which neutrophils remain alive and continue to crawl. This mechanism was observed in vivo during *Staphylococcus aureus* infections, preventing bacterial dissemination [48]. NETosis and NET release mechanisms vary according to the trigger [53]. NETosis depends on the influx of Ca^2+^ into activated neutrophils. This relies on multiple pathways including NOX activation, protein kinase as Protein Kinase C (PKC), Raf-MEL-ERK MAP, or SYK-PI3K pathways, as well as the recently discovered JAK2 (for reviews: [42,53,54]). However, further studies are required to identify the original mediator that explain the cellular fate in chromatin decondensation and DNA excretion. Notably, the executioner protein-mediating NETosis has been recently suggested to be the protein GSDMD [55,56], which was initially discovered as a pyroptosis executioner in macrophages [10,11]. Interestingly, other cell types such as eosinophils, basophils, mast cells, and macrophages also mount a NETosis-like process [53]. However, whether NETosis mechanisms are similar between cell lineages remains unknown. The functional relevance of this process beyond the neutrophil biology also needs further investigations.

Over the last decade, neutrophils emerged as a more complex cell type that initially thought endowed with a wide set of immunomodulatory properties through their ability to produce various cytokines, chemokines, and growth factors as well as mediating functions through cell-cell interactions. Thus, neutrophils exert roles during cancer, infection, and inflammatory disorders by regulating intensity and quality of the ensuing immune response, mediating antibody response and influencing tissue repair and angiogenesis [57,58,59,60]. It remains to be determined whether these versatile properties are dependent on functional plasticity or are mediated by specific sublineages. In line, increasing evidence suggests the existence of neutrophilic subsets with discrete functions. For instance, circulating neutrophils can be segregated based on their density [61,62]. Low-density neutrophils (LDN) have been observed in various pathologies including systemic lupus erythematosus [63], cancer [64], tuberculosis [65], malaria [66], or asthma [67]. In these settings, LDN appeared to display immunosuppressive functions [68]. Myeloid-derived suppressor cells (MDSC) represent a heterogeneous population of immature cells that comprise monocytic MDSC (M-MDSC) and granulocytic MDSC (Polymorphonuclear neutrophils (PMN)-MDSC). PMN-MDSC are phenotypically similar to mature neutrophils [69] but studies proposed that they expressed higher levels of CD115 (CSF1R), CD244 (2B4), and lectin-type oxidized LDL receptor 1 [70,71]). Initially described in cancer, PMN-MDSC have also been identified during bacterial, fungal, and viral infections [72]. Similar to LDN, PMN-MDSC can accumulate under pathological conditions and exhibit immunosuppressive activities via different mechanisms, including production of reactive oxygen species (ROS), nitric oxide (NO), prostaglandin E2 (PGE2), or arginase 1 [73]. Akin to macrophages, tumor-associated neutrophils (TAN) have been identified and can be segregated in two subsets according to their activation profile, cytokine signature, and effects on tumor progression. While N1 TAN exert antitumoral activity by direct or indirect cytotoxicity, N2 TAN promote tumor growth via immunosuppression as well as increased angiogenesis and metastasis [74]. Furthermore, as compared with classical neutrophils and PMN-MSDCs, N2 TAN produce higher levels of chemokines but lower granules and ROS [75,76]. PD-L1-expressing neutrophils have been identified in cancer, infections, autoimmunity, and sepsis [77,78,79,80,81] and appear to be less prone to apoptosis than their PDL-1-negative counterparts [81]. Interestingly, these PDL-1-expressing neutrophils suppress the immune response by limiting activity of PD-1-expressing cells [77,78,79,80,81], culminating in either a protective or deleterious role according to the pathological context.

The neutrophils are no longer the simple cell patrol, catching and eliminating pathogens, but they have multiple innate immune mechanisms in response to various conditions, including infections, autoimmune disorders, and cancers. Therefore, neutrophils clearly emerged as crucial components in regulating the immune responses.

## 3. Neutrophilic NLRP3 Inflammasome: A Novel Relevant Regulatory Pathway

### 3.1. Emerging Shreds of Evidence

Although inflammasomes were historically thought to be selectively activated by cell populations from the myeloid lineage (e.g., monocytes, macrophages, and dendritic cells) [5], NLRP3 inflammasome components were subsequently widely detected in both hematopoietic and non-hematopoietic cells, including neutrophils [82,83]. Indeed, the functional relevance of the inflammasome in neutrophils has first received minor consideration since pioneer studies suggested that the processing of mature forms of IL-1β by neutrophils was caspase-1 independent, and required the processing from neutrophil serine proteases such as proteinase 3 and elastase [84,85,86]. A few years later, a paradigm shift identified NLRP3-dependent IL-1β secretion in vitro by both highly purified mouse and human neutrophils [87,88]. However, the amounts of NLRP3-dependent IL-1β released by neutrophils were relatively low as compared with macrophages, which raised some doubts on the physiological relevance of NLRP3 inflammasome in neutrophils. However, during immune responses, neutrophils are massively recruited to the site of infection and inflammation, which may compensate for the limited amount of IL-1β produced per neutrophil as compared with macrophages. Moreover, a recent study suggested that neutrophils could release IL-1β faster than macrophages [89].

In line, a seminal study by Cho and colleagues provided the first in vivo evidence for a role of the NLRP3 inflammasome in IL-1β secretion by neutrophils in a model of *Staphylococcus aureus* acute infection [90]. However, this study relied on the use of chemical inhibitors (e.g., glyburide and Z-YVAD-FMK) that may lack complete specificity and induce side effects [91]. A step forward into our understanding of the functional relevance of the neutrophilic NLRP3 inflammasome arose from Pearlman’s lab [92,93], who elegantly demonstrated using gene-targeted mice and in vivo depletion that neutrophils represented a prime source of IL-1β in a NLRP3-, ASC- and caspase-1/11-dependent manner during *Streptococcus pneumoniae* (serotype 4) corneal infection. In addition, analysis of protein expression of NLRP3 inflammasome components from purified bone marrow neutrophils incubated with heat-killed *S. pneumoniae* indicated NLRP3 priming and activation [93]. Furthermore, the use of neutrophils from elastase-deficient mice confirmed that serine proteases were not required for IL-1β processing in this infection model [93]. Concomitantly, our lab reinforced the existence of a fully functional NLRP3 inflammasome in neutrophils in a model of *S. pneumoniae*-induced pneumonia [94]. Western blotting on ex vivo purified primary lung neutrophils from *S. pneumoniae*-infected mice confirmed NLRP3 priming and activation [94], as previously observed in in vitro assays [93]. We also demonstrated that human neutrophils from peripheral blood mononuclear cells of healthy donors produced IL-1β in an NLRP3-dependent manner in response to *S. pneumoniae* (serotype 1) using MCC950 [94], a selective NLRP3 inflammasome inhibitor [95].

Despite these studies point toward a role for the neutrophilic NLRP3 inflammasome in vivo, they still present limitations mainly due to the lack of selective deletion of NLRP3 components in neutrophils. However, this drawback could be overcome in the near future thanks to several conditional mouse models that specifically target neutrophils such as the *Mrp8-Cre* mice [96] and the Catchup mice (by targeting the locus Ly6G with a knock-in allele expressing the Cre recombinase and fluorescent protein) [97]. Interestingly, a mouse model with specific *Nlrp3* gain-of-function mutations in neutrophils was generated [98]. Thus, the development of novel mouse models based on these existing biological tools will likely generate meaningful data to better appreciate the biological relevance of the inflammasomes in neutrophils. 

For a long time, neutrophils have been considered to be unable to assemble inflammasome, but today the compiling proofs withdraw the doubt and confirm that neutrophils should be highly regarded in the field of the inflammasome.

### 3.2. Activation Mechanisms

Akin to other myeloid cells, NLRP3 inflammasome in neutrophils has been proposed to be activated via multiple triggers using canonical and non-canonical pathways.

#### 3.2.1. Canonical Pathway

The canonical pathway is efficiently engaged in neutrophils as in other myeloid cells, despite a few differences (Figure 2).

##### Priming Step

Pioneer experiments by Hartl’s and Hornung’s groups revealed that both mouse and human neutrophils could be primed for IL-1β production in vitro with purified LPS resulting in increased transcription of *Il1b* and *Nlrp3* mRNA [87,88]. Next, mechanisms driving NLRP3 inflammasome priming in neutrophils in response to pathogenic bacteria were investigated. For instance, the Gram-negative bacterium uropathogenic *Escherichia coli* [99] induced NLRP3 inflammasome priming in human neutrophils independently of TLR signaling [100]. A TLR-independent pathway was also observed in the context of infection with the Gram-negative bacterium *Helicobacter pylori* using blocking antibodies targeting TLR2 and TLR4 (Figure 2A) [101]. However, a subsequent study suggested that *H. pylori* could prime neutrophils in a TLR2-dependent (but TLR4- and NOD2-independent) manner using purified cells from gene-targeted mice (Figure 2A) [102]. Since the same bacterial strain was used in both studies, it is possible that these discrepant results arose from the different cellular sources used (lines vs. primary; human vs. mouse). Thus, the mechanisms involved in neutrophilic NLRP3 inflammasome priming in response to Gram-negative bacteria require additional studies.

Mechanistic studies have also been conducted using the Gram-positive bacteria *Staphylococcus aureus* and *Streptococcus pneumoniae*, two pathogens with an extracellular lifestyle that account for the most severe infections at barrier sites. In the presence of live *S. aureus* mouse bone marrow neutrophils produced IL-1β that was dependent on multiple PRRs including TLR2, NOD2 and formyl peptide receptor 1 (FPR-1), suggesting the contribution of these different innate sensors in NLRP3 inflammasome priming (Figure 2A) [90]. In response to *S. pneumoniae*, multiple pathways leading to inflammasome priming in neutrophils have been proposed [93,94]. In vitro exposure of mouse bone marrow neutrophils with heat-killed (HK) *S. pneumoniae* resulted in a TLR2-dependent NLRP3 upregulation (Figure 2A) [93]. Moreover, the use of pharmacological inhibitors indicated that TLR2-dependent NLRP3 priming in neutrophils occurred through the NF-κB and MAPK/AP-1 pathways [93]. Alternatively, our lab observed that NLRP3 inflammasome priming in lung neutrophils during *S. pneumoniae*-induced pneumonia was dominated by host-derived TNF-α secreted by alveolar macrophages (Figure 2A) [94], although TNF-α has been shown to exhibit weaker and delayed ability to prime NLRP3 inflammasome in macrophages as compared with LPS [103]. However, we observed that TNF-α could prime the NLRP3 inflammasome in neutrophils as early as 6 h post-*S. pneumoniae* infection [94]. A TNF-dependent NLRP3 priming in neutrophils appears to also operate during sterile inflammation [104,105]. Thus, it is possible that the role of TNF-α in neutrophilic inflammasome priming dominates as compared with its contribution in macrophages. In line, neutrophils express higher transcripts of *Tnfrsf1a* (encoding for TNFR1) as compared with macrophages [106,107].

Since neutrophils can prime NLRP3 inflammasome through bacterial- and host-derived molecules, the host is able to engage innate immunity upon direct or indirect contact with bacteria whatever the site of infection.

##### Activation Step

In macrophages, the mechanisms leading to NLRP3 inflammasome oligomerization are multiple, and a similar diversity of stimuli seems to be at play in neutrophils.

Classical activators such as extracellular ATP and nigericin—that both mediate K^+^ efflux in vitro—activate NLRP3 inflammasomes in mouse and human neutrophils (Figure 2A) [87,88]. During bacterial infection, K^+^ efflux appears as a main driver in NLRP3 activation in neutrophils, although the triggering mechanisms differ among studies. For instance, pore-forming toxins from *Escherichia coli* (e.g., α-hemolysin), *Staphylococcus aureus* (e.g., enterotoxin O), *Streptococcus agalactiae* (e.g., β-hemolysin), and *Streptococcus aureus* (e.g., pneumolysin) lead to an increase in K^+^ efflux in neutrophils and subsequent NLRP3 inflammasome assembly (Figure 2A) [93,94,99,108,109]. Additionally, during *S. pneumoniae* corneal infection, the extracellular release of ATP also induced K^+^ efflux in P2X receptor 7 (P2X7R)-expressing neutrophils, culminating in NLRP3 inflammasome oligomerization (Figure 2A) [92]. Exogenous ATP potentiates NLRP3 activity in neutrophils in response to the enterotoxin O from *S. aureus*, although the biological relevance in vivo remains to be determined (Figure 2A) [109].

Insoluble particulate and crystalline compounds are efficient activators of NLRP3 inflammasome in macrophages and dendritic cells [110]. This activation is mediated by phagolysosomal disruption leading to the cathepsin-dependent assembly of the NLRP3 inflammasome [111]. Schroder’s lab has suggested that in neutrophils, the particles or crystals weakly—if not—activated NLRP3 inflammasome [112]. However, two other studies have challenged this observation and demonstrated that silica and uric acid may trigger NLRP3 inflammasome activation in neutrophils [104,113]. These contradictory observations are likely due to the different origin of neutrophils. Interestingly, serum amyloid A could induce NLRP3 inflammasome activation in neutrophils independently from K^+^ efflux [114] and be partially dependent on the spleen tyrosine kinase Syk [115].

Regarding the existing literature, the mechanisms of canonical NLRP3 inflammasome activation appeared to be similar to the one observed for the macrophages or monocyte. However, it remains to be determined whether other general NLRP3 inflammasome assemblies and activation mechanisms described in macrophages are functional in neutrophils, including mitochondrial- or reticulum endoplasmic-dependent pathways. 

#### 3.2.2. Non-Canonical and Alternative Pathways

The non-canonical NLRP3 inflammasome represents an additional layer of defense during infection and is mainly activated in response to intracellular pathogens (Figure 2B). Caspase-11 senses the cytosolic LPS release by the intracellular bacteria and mediates inflammasome activation (Figure 2B) [26]. Neutrophils transfected with LPS activate the non-canonical inflammasome in caspase-11 and caspase-1 in the same manner as in macrophages (Figure 2B) [55,116]. However, no pyroptosis was observed despite GSDMD cleavage (Figure 2C), but rather a reliance on NETosis program cell death with chromatin extrusion (Figure 3) [55]. GSDMD processing is entirely dependent on caspase-11 [55,116]. During *Burkolderia thailandensis* infection, non-canonical inflammasome is activated by both caspase-1 and -11 [116]. However, while IL-1β processing depended on caspase-1, GSDMD cleavage, neutrophil cell death, and invader pathogen clearance were mediated by caspase-11 (Figure 2B and Figure 3) [116]. The fungus *Aspergillus fumigatus* also engages the non-canonical inflammasome pathway in neutrophils [117], leading to caspase-11-dependent activation of caspase-1 for IL-1β processing (Figure 2B). Caspase-11 is regulated by type I IFN receptor-dependent JAK-STAT signaling. Using swollen conidia, upregulation of the IL-1β pro-form is controlled by the Dectin-1-dependent Raf1 pathway (Figure 2B) [117]. Interestingly, no signs of function for caspase-11 was observed in vitro with macrophages; however, the mice lacking caspase-11 were susceptible to invasive pulmonary aspergillosis [118,119]. The underlying mechanisms for caspase-11 requirement during fungal infection still need to be elucidated.

Recently, the priming has been shown to be dispensable for NLRP3 inflammasome activation in human neutrophils as for human monocyte [120,121]. This new observation highlights the difference observed between the human and mouse models, and the importance of studying both models to decipher the underlying molecular mechanism. However, the mechanism driving this alternative inflammasome activation is not known yet, which needs further exploration [120].

Granulocyte-macrophage colony-stimulating factor (GM-CSF) is sufficient to induce NLRP3-dependent secretion of IL-1β by human neutrophils [122]. In vitro treatment of neutrophils with recombinant GM-CSF led to upregulation of *Il1b* and *Nlrp3* transcripts in a JAK/STAT-dependent manner [122] as previously shown in macrophages [123]. However, it remains unclear how GM-CSF controls NLRP3 inflammasome assembly and activation, and further studies are warranted.

### 3.3. Does NLRP3 Inflammasome-Mediated Cell Death Occur in Neutrophils?

An essential function of the inflammasome is to sense the intracellular pathogen to maintain tissue homeostasis via two main steps. First of all, the inflammasome releases the pro-inflammatory cytokine IL-1β to mediate immune cell recruitment to the site of infection. Then, inflammasome induces death of the infected cells to control pathogen dissemination [27]. Early studies on the inflammasome functions in neutrophils showed that the inflammasome components (NLRP3, ASC, caspase-1) were expressed, and released IL-1β, but surprisingly in absence of pyroptosis (Figure 2C) [55,92,93,124]. The mechanisms associated with pyroptosis resistance in neutrophils remain unclear. Emerging evidence suggest a fine-tuning of GSDMD cleavage and pore-forming at the plasma membrane [27]. It is proposed that the neutrophils have a low expression level of ASC and caspase-1 as compared with macrophages, resulting in weak inflammasome complex formation [55,125], which is essential for caspase-1 activity and duration [126].

Interestingly, caspase-1 is more efficient in processing the pro-IL-1β than GSDMD protein, which may probably maintain neutrophils in a sublytic state allowing for IL-1β release without cell death [55], as was observed in macrophages (Figure 2C) [127,128]. However, the formation of GSDMD-dependent pores of the plasma membrane is still unknown and debated. 

A study from Pearlman’s lab discovered that the GSDMD^NT^ domain generated during NLRP3 inflammasome signaling in neutrophils localizes within the membranes of azurophilic granules and intracellular organelles rather than the plasma membrane (Figure 3) [129]. However, the mechanisms accompanying the intracellular trafficking of GSDMD are unknown and require further investigation. Carty et al. recently demonstrated that sterile alpha and toll/interleukine-1 receptor motif-containing protein 1 (SARM1) is involved in pyroptosis during NLRP3 inflammasome activation (Figure 2C) [130]. Macrophages lacking SARM1 have increased inflammasome activation, including caspase-1 cleavage, ASC speck formation, and IL-1β release (Figure 2C). However, pyroptosis is almost abolished, whereas GSDMD is still processed. SARM1 induces mitochondria depolarization that mediates cell death, but the fine mechanism is still unknown [130]. Although this study confirmed the crucial role of mitochondria in regulating NLRP3 inflammasome, it raised two questions: does SARM1 participate in GSDMD^NT^ trafficking to the plasma membrane for efficient pyroptosis? Do other intrinsic or extrinsic TIR domain proteins regulate pyroptosis? Interestingly, neutrophils express lower levels of SARM1 than macrophages, which may partially explain the absence of pyroptosis in neutrophils (Figure 2C) [89]. Mitochondria are essential in fueling the cells and cell death fate decision (apoptosis). Mitochondria in neutrophils display discrete metabolic properties [131]. Neutrophilic mitochondria do not synthesize ATP and have limited enzymatic activity [131]. So, is the lack of pyroptosis in neutrophils due to mitochondria dysfunction? Studying the mitochondrial metabolism in neutrophils vs. macrophages may reveal new mechanisms in pyroptosis regulation.

Intriguingly, while being unable to mediate pyroptosis in neutrophils, GSDMD controls NETosis and NETs formation upon LPS-mediated non-canonical NLRP3 inflammasome and Gram-negative bacteria infection (Figure 3) [55,132]. Indeed, caspase-11 cleaves GSDMD more efficiently than caspase-1 and that may explain why cell death in neutrophils could be better controlled by the non-canonical pathway over the canonical NLRP3 inflammasome (Figure 2 and Figure 3) [55]. GSDMD-mediated NETosis is independent of neutrophil elastase, myeloperoxidase, and PAD4, which are key regulators of classical NETosis [55]. SARM1 seems to be dispensable in GSDMD-mediated NETosis, nuclear delobulation, DNA extrusion, and rupture of the nuclear membrane. SARM1 may be specific to pyroptosis by controlling GSDMD pore formation at the plasma membrane. Thus, it will be of interest to investigate the trafficking of GSDMD and other gasdermin family members in neutrophils and macrophages upon multiple cell death triggers.

In term, neutrophils have a short lifetime, but they are surprisingly resistant to pyroptosis during inflammasome activation due to a GSDMD pore-forming fine-tuning. However, non-canonical NLRP3 inflammasome activation mediates a complex lytic cell death called NETosis with nuclear extrusion and DNA net release.

**Figure 3 cells-11-01188-f003:**
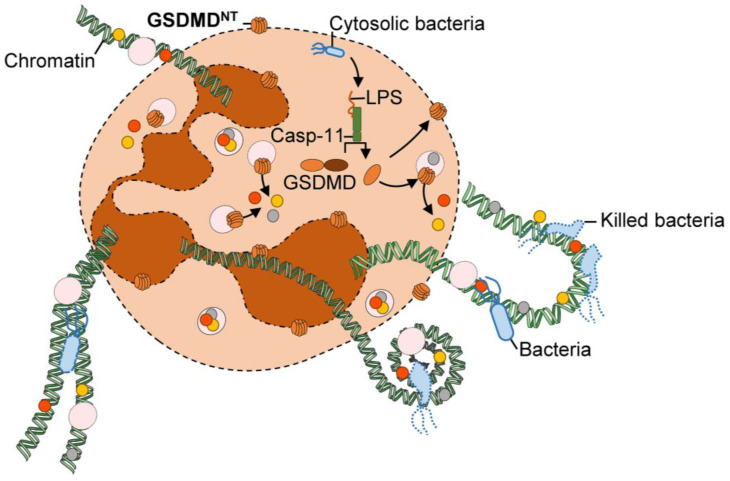
Mechanism of GSDMD-mediated NETosis. The LPS released intracellularly by Gram-negative bacteria binds to the protease caspase-11. In turn, activated caspase-11 highly efficiently cleaves Gasdermin D (GSDMD) which releases GSDMD^NT^ into the cytoplasm of neutrophils. The Gasdermin D N-terminal domain (GSDMD^NT^) lytic fragment is localized within the membranes of azurophilic granules and intracellular organelles rather than the plasma membrane of neutrophils. The pore formation in neutrophil organelles enables the release of enzymes and proteases that may participate in nuclear delobulation, histone citrullination, DNA extrusion, and rupture of the nuclear envelope. The activated cells died under the neutrophil-specific program cell death mediating the formation of NET (neutrophil extracellular traps) named as NETosis. Then, NETs are released in the milieu to capture and to kill the bacteria.

### 3.4. Relevant Functions of the NLRP3 Inflammasome Activation by Neutrophils in Diseases

The biological relevance of the neutrophilic NLRP3 inflammasome has been evaluated in various conditions, including sterile inflammation and infections conferring to this pathway, either protective or deleterious phenotypes. Its role was mainly investigated by combining data from NLRP3 inflammasome component-deficient and neutrophil-depleted mice. However, the recent generation of mouse models with specific deletion of NLRP3 inflammasome components in neutrophils enabled a more thorough evaluation of this pathway. 

#### 3.4.1. (Auto)-Inflammation

Cryopyrin-associated periodic syndrome (CAPS) represents a group of heterogeneous autoinflammatory disorders caused by gain-of-function (GOF) mutations in NLRP3 [133]. This leads to constitutive production of IL-1β and culminates in systemic inflammation with multiple clinical manifestations including fever, skin rash, joint inflammation, and keratitis [134]. Using two mouse models displaying GOF mutations in NLRP3 (*Nlrp3*^A350V^ and *Nlrp3*^L351P^) associated with two clinical entities of CAPS, namely Muckle-Wells syndrome and familial cold autoinflammatory syndrome [135], authors demonstrated that targeted and specific GOF mutations in NLRP3 in neutrophils (*Nlrp3*^A350V^*MRP8^cre^* and *Nlrp3*^L351P^*MRP8^cre^* mice model) were sufficient to trigger severe CAPS-like phenotype as judged by the intense extent of skin inflammation [98]. Indeed, the skin-infiltrating neutrophils of patients with CAPS syndrome represent an essential source of bioactive IL-1β [98]. Regarding skin-infiltrating macrophages and dendritic cells, their contribution was partial, as only 10% of these cells released IL-1β, whereas the CAPS syndrome was totally independent of mast cells [98]. This study strongly suggests that neutrophils represent the leading cause of CAPS pathologies and should be considered as specific targets to modulate IL1β release. Therefore, a bispecific antibody targeting neutrophils and the IL-1 pathway would be a good therapeutic tool.

Gouty arthritis is a debilitating and painful inflammatory condition caused by the deposition of monosodium urate crystals in joints [136]. In an experimental model of gout in rat, neutrophilic NLRP3 inflammasome has been suggested as an essential source of IL-1β contributing to enhancing inflammation and pain [113]. However, the mechanisms involved in NLRP3 activity have not yet been investigated. Regarding the limited capacity of neutrophils to activate NLRP3 inflammasome through crystalline compounds, the activity of the neutrophilic NLRP3 inflammasome may rely on classical inflammation-associated host-derived molecules such as alarmins (e.g., DNA, HSP, HMGB1, Ca^2+^-binding S100a8/a9 proteins, ATP or hyaluronic acid). Interestingly, the ketogenic diet (β-hydroxybutyrate) in the gouty arthritis model rat model can inhibit both NLRP3 inflammasome priming and assembly step in neutrophils [113]. 

#### 3.4.2. Infections

During *Staphylococcus aureus* cutaneous infection, abscess formation is a mandatory step in developing the host immune response, a process that largely relies on neutrophil recruitment. Interestingly, NLRP3 inflammasome-dependent IL-1β release by neutrophils appeared to be essential in this process, which conferred a role in bacterial clearance [90]. The neutrophilic NLRP3 inflammasome was also proposed to contribute to the host response in various models of *Streptococcus pneumoniae* infections [92,93,94]. In an *S. pneumoniae* corneal infection model, NLRP3-dependent IL-1β secretion by neutrophils controlled their own recruitment to facilitate bacterial clearance at the expense of increased keratitis [92,93]. During pneumococcal respiratory infection, neutrophilic NLRP3 inflammasome-dependent IL-1β secretion also controlled neutrophil recruitment into the lung tissues to exert their antimicrobial properties [94]. However, in this model, we demonstrated that IL-1β indirectly recruited neutrophils by activating IL-17A-producing γδT cells [94].

A role for neutrophil-derived IL-1β has also been demonstrated during corneal infection with the filamentous fungus *A. fumigatus* [117]. Similar to the *S. pneumoniae* model [93], neutrophilic NLRP3-dependent IL-1β secretion controlled neutrophil recruitment to contain the hyphal mass at the expense of more corneal opacity [117].

The contribution of the NLRP3-inflammasome in neutrophils during viral infection has recently emerged during COVID-19. Both alterations in functions of neutrophils and the NLRP3 inflammasome appear to contribute to COVID-19 pathogenesis [137,138]. Interestingly, a recent clinical study revealed that circulating neutrophil subsets from COVID-19 patients display reduced intrinsic caspase-1 activity in response to classical NLRP3 inflammasome activators [138]. Of note, decreased activity was accentuated during severe clinical episode and returned to normal levels—as observed in healthy donors—when patients recovered [138]. Although the underlying mechanisms are unknown, this may be linked to the increased proportion of “immature/suppressive” neutrophils observed in severe forms of COVID-19. Alternatively, a negative regulatory mechanism may dampen the inflammasome activation [137,139]. However, in this study, the authors triggered NLRP3 inflammasome activation in circulating neutrophils with nigericin only [138]. It will be of high interest to test other NLRP3 inflammasome triggers to understand whether the mechanism specifically depends on the pore forming toxins. Moreover, as circulating neutrophils may differ from their extravasated counterparts, the role of inflammasome in airway neutrophils during COVID-19 remains to be clarified. However, circulating and airway neutrophils from severe COVID-19 patients were recently shown to have already ASC speck formed and release active IL-1β and IL-18 cytokines with positive citrullinated Histone 3 nuclei, suggesting a link between inflammasome and NETosis [120].

Interestingly, the Marichal’s lab observed that the pathological function of neutrophils in severe COVID-19 infection was due to the release of NETs both in lung airway tissue and the vascular compartment [47]. The NETs formation is a marker for severe pulmonary complications in COVID-19. However, the link between the NLRP3 inflammasome and NETs release has not yet been explored in this setting. The investigation of GSDMD cleavage in airway-infiltrating neutrophils should reveal interesting mechanisms and open the way for new treatment options in severe pneumonia.

A reduced activity of the neutrophilic NLRP3 inflammasome during infection was also illustrated in vitro in response to the protozoan *Toxoplasma gondii* infection [140]. *T. gondii* inhibited phosphorylation of the p65 subunit of NF-κB in LPS-primed neutrophils affecting the NLRP3 inflammasome priming step [140]. Thus, this mechanism may represent a pathway for *T. gondii* to subvert neutrophil-dependent immunity.

Altogether, neutrophilic NLRP3 inflammasome emerged as a key pathway in orchestrating the immune response in many pathological conditions. Further studies will be essential to evaluate its contribution to low-grade inflammatory conditions such as cancer and/or chronic infections. Of note, NLRP3 inflammasome in neutrophils has been shown to be activated during *H. pylori* [102]. Since IL-1β is a critical factor in the development of gastrointestinal cancer during chronic *H. pylori* infection [141], this pathway should also be explored in this pathophysiological process. 

## 4. Concluding Remarks

After long debate, doubt on inflammasome activity in neutrophils has now been dismissed. Neutrophils can effectively activate the canonical and non-canonical NLRP3 inflammasomes as macrophages (Table 1). However, many gaps still need to be filled. The regulatory mechanisms of NLRP3 are complex, involving an array of transcription factors and regulatory molecules, which need to be fine-tuned [9]. The requirement of Nina-related kinase 7 (NEK7)-binding on NLRP3 inflammasome in neutrophils is still unknown, and only a few NLRP3 triggers have been tested and identified. The knowledge of activators or triggers, as well as the identification of host signaling components, needs to be expanded. Large screens using CRISPR technology may be valuable to understand inflammasome regulation in neutrophils. Since the genetic manipulation of primary neutrophils is almost impossible, the future generation of neutrophil cell lines will be likely helpful. 

Only a few pathogens are described to regulate inflammasome in neutrophils. *A. fumigatus* activates caspase-11 in neutrophils [117], whereas in macrophages, inflammasome activation is caspase-11 independent [119]. Since fungi do not produce LPS, ligands mediating caspase-11 activation need to be identified. Recently, the galactosaminogalactan (GAG) of *A. fumigatus* has been shown to mediate NLRP3 inflammasome activation by reticulum endoplasmic stress mechanism [25]. Do GAG or other fungal polysaccharides bind to caspase-11 to activate NLRP3 in neutrophils?

The cell-specific regulation of death in neutrophils is fascinating. Neutrophils were long-considered short-lived, but they can expand their life time under inflammatory conditions by switching off apoptosis. Neutrophils are fully resistant to pyroptosis, despite clear signs of GSDMD activation. The activation of GSDMD and release of the lytic executioner GSDMD^NT^ forms pores in the intracellular vesicle compartment but not at the membrane, which induces NETosis-like cell death. Thus, the regulation of pyroptosis in neutrophils is not as straightforward as defined in macrophages. Additional efforts on the understanding of cellular trafficking of GSDMD^NT^ are needed and may reveal new and exciting pathways. The roles of other members of the GSDM family (GSDMA, GSDMB, GSDMC, GSDME, and GSDMF) in neutrophil biology are currently unknown. For instance, GSDME plays a crucial role in specific cell type and caspase-3/-7 apoptosis [142,143]. Do other GSDM family members regulate NETosis cell death and NETs release?

For obvious technical reasons, many studies are performed ex vivo using neutrophils isolated from mouse bone marrow or human blood. However, in vivo functions of inflammasomes in neutrophils during infection or autoinflammatory disease are likely more complex. The recent development of mice with conditional NLRP3 inflammasome deficiency in neutrophils represents an important step forward. Such tools revealed that the pathological GOF in NLRP3 during CAPS was dependent on neutrophils [98]. In the near future, generation of mice with specific deletion of given inflammasome components in neutrophils (*Nlrp3^fl/fl^MRP8^cre^*, *Asc^fl/fl^MRP8^cre^ Caspase-1^fl/fl^MRP8^cre^*, *Caspase-11^fl/fl^MRP8^cre^*, *GSDMD^fl/fl^MRP8^cre^*) will be highly valuable. Recently, *Caspase-1^fl/fl^MRP8^cre^* mice enabled demonstrating that neutrophils activated NLRC4 inflammasome in response to *P. aeruginosa* and were competent for pyroptosis activation upon “incomplete NETosis” [144]. This interesting study raises key questions: is this mechanism specific to *P. aeruginosa* infection and/or neutrophilic NLRC4 inflammasome? In any case, this study nicely illustrated the complex conundrum of cell death regulation within neutrophils and the importance of further dedicated studies.

Here, we reviewed and discussed the activation mechanisms of NLRP3 inflammasome in neutrophils as well as its functional relevance during infections and inflammatory diseases. Although our understanding of the neutrophilic NLRP3 inflammasome biology is still in its infancy, its specific harnessing may represent a relevant option for new cell-based immunotherapies. In addition, neutrophils appear to have the potential to activate other inflammasomes such as NLRC4 and Aim2 even though the regulatory mechanisms involved are unclear. Altogether, neutrophils now emerge as critical cellular components in the biology of inflammasomes. Further investigations should be encouraged to better appreciate the fine mechanisms of action regulation and clinical relevance beyond infection and inflammation.

## Figures and Tables

**Figure 1 cells-11-01188-f001:**
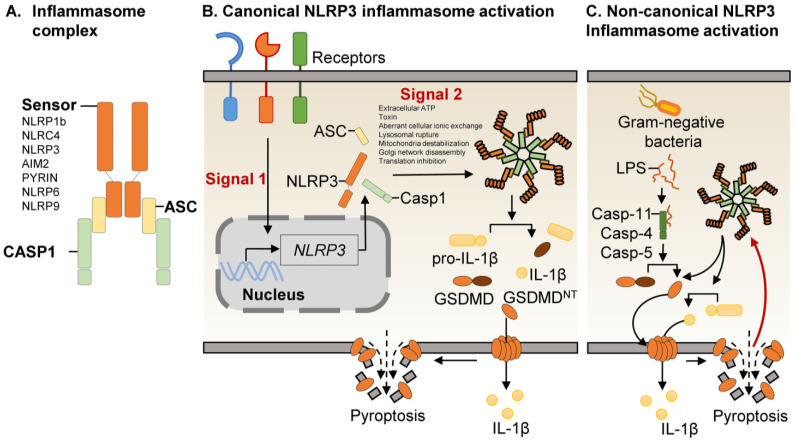
NLRP3 inflammasome assembly. (**A**). Inflammasome complex. Inflammasomes are composed of cytosolic sensors, which trigger caspase-1 activation. The identified cytosolic sensors are the NLR family pyrin domain-containing 1b (NLRP1b), NLR family CARD domain-containing protein 4 (NLRC4), and NLR family pyrin domain-containing 3 (NLRP3), absent in melanoma-2 (AIM2) receptor and pyrin receptor (PYRIN), NLR family pyrin domain-containing 6 (NLRP6) and NLR family pyrin domain-containing 9 (NLRP9). First, the sensors recruit the adaptor apoptosis-associated speck-like protein containing a CARD (ASC) via domain-domain (PYRIN-PYRIN) interaction. Then, ASC recruits and interacts with the caspase-1 CARD domain by domain-domain interaction, resulting in the assembly of a functionally mature inflammasome. (**B**). Canonical NLRP3 inflammasome activation. The canonical NLRP3 inflammasome requires two parallel and complementary steps: (1) priming (signal 1) after sensing invaders or sterile insult, which induces the transcription of NLRP3 inflammasome components (NLRP3, pro-IL1β, …) and (2) activation, which results in the assembly of the NLRP3 inflammasome, Gasdermin D (GSDMD)-dependent pore formation, pyroptosis, and IL-1β release. (**C**). Non-canonical NLRP3 inflammasome activation. The non-canonical NLRP3 inflammasome is engaged in response to Gram-negative bacteria by the binding of LPS on the protease caspase-11 (mouse) or caspase-4/-5 (human). Activated caspase-11 or caspase-4/-5 cleaves GSDMD and induces pore formation, potassium efflux, culminating in NLRP3 inflammasome activation.

**Figure 2 cells-11-01188-f002:**
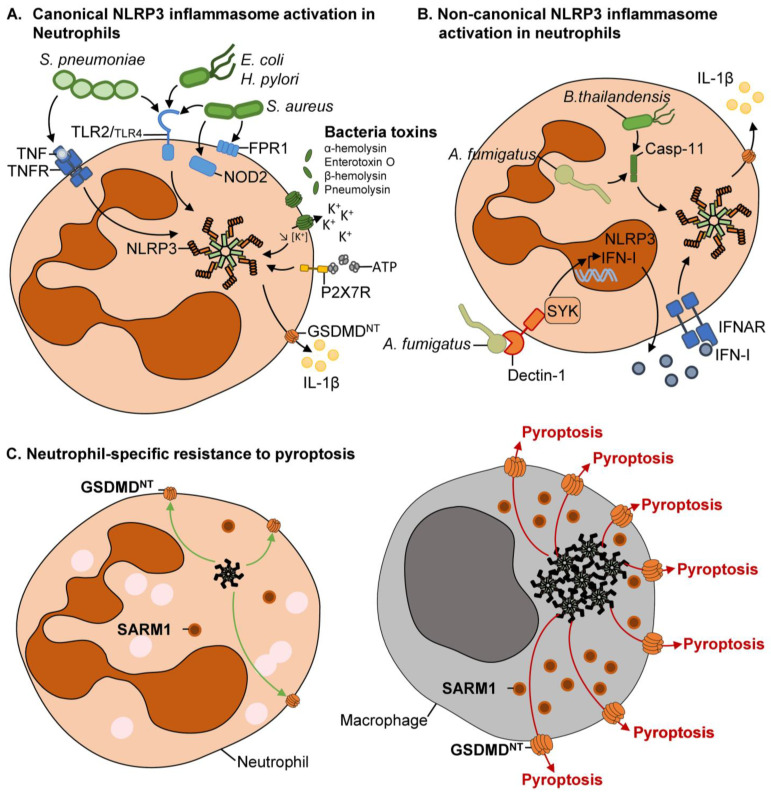
Mechanism NLRP3 inflammasome activation in neutrophils. (**A**). Canonical NLRP3 inflammasome activation in neutrophils. Extracellular bacteria are mainly sensed by Toll-like receptor 2 (TLR2) and, in some cases, TLR4, Nucleotide-binding oligomerization domain-containing protein 2 (NOD2) or formyl peptide receptor 1 (FPR1), leading to efficient priming. This first step can also be mediated indirectly by host factors such as TNF and TNFR engagement. The main NLRP3 inflammasome activation signals are dependent of bacterial toxins from *Escherichia coli* (e.g., α-hemolysin), *Staphylococcus aureus* (e.g., enterotoxin O), *Streptococcus agalactiae* (e.g., β-hemolysin), and *Streptococcus aureus* (e.g., pneumolysin). The damages they cause to the neutrophil lead to increased potassium (K^+^) efflux, extracellular ATP, and subsequent NLRP3 inflammasome assembly via P2X receptor 7 (P2X7R). (**B**). Non-canonical NLRP3 inflammasome activation in neutrophils. The non-canonical NLRP3 inflammasome is activated in response to Gram-negative bacteria (*Burkholderia thailandensis*) or filamentous fungi *Aspergillus fumigatus.* This triggers caspase-11 activation and then NLRP3 inflammasome activation. In response to *A. fumigatus* this pathway depends on Dectin-1, spleen tyrosine kinase (SYK), and type I interferon (IFN-I) signaling. (**C**). Neutrophil-specific resistance to pyroptosis. Neutrophils are resistant to pyroptosis likely because they produce low ASC, caspase-1, and sterile alpha and TIR motif-containing 1 (SARM1) proteins as compared with macrophages. Indeed, SARM1 seems to be a key regulator of the rate of Gasdermin D (GSDMD) pore formation at the plasma membrane to mediate pyroptosis. Thus, low SARM1 production by neutrophils may lead to decreased cell lysis.

**Table 1 cells-11-01188-t001:** Specific and conserved activation mechanisms of NLRP3 inflammasome and associated-cell death in macrophages vs. neutrophils. The table depicts only conditions in which mechanisms between macrophages and neutrophils have been compared. TLR, Toll-like receptors; TNF-R, tumor necrosis factor α receptor; IL-1R, interleukine-1 receptor; NOD, nucleotide-binding oligomerization domain; CLR, C-type lectin receptors; NF-κB, Nuclear factor kappa B; MAPK, mitogen-activated protein kinase; ASK, apoptosis signal-regulating kinase; CASP8, caspase-8; FADD, FAS-associated death domain; AP-1, Activator protein 1; Syk, spleen tyrosine kinase; K, potassium; GSDMD, gasdermin D; CASP11, caspase-11; LPS, lipopolysaccharide; GBP, guanylate-binding proteins; IRGB10, immunity-related GTPase family member b10; IFN, interferon; NETosis, neutrophil extracellular traps-osis.

Pathway	Steps		Macrophages	Neutrophils
Canonical pathway	Priming	Known sensors	TLRs, TNF-R, IL-1R, NOD1/2, CLRs	TLRs, TNF-R, IL-1R, NOD2, CLRs
		Known activated pathways	NF-κB, MAPK, ASK, CASP8/FADD and AP-1	NF-κB
	Activation		K^+^ efflux induced by pore-forming toxins and extracellular ATP - Particles and crystals (strong activation) - Bacterial infection - Dectin-1/Syk activation induced by fungal infection and fungal PAMPs	K^+^ efflux induced by pore-forming toxins and extracellular ATP - Particles and crystals (weak activation) - Bacterial infection - Syk activation induced by serum amyloid A
	NLRP3-dependent cytokine secretion (on a per cell basis)		+++	+
	Cell death		GSDMD-dependent pyroptosis	No pyroptosis
Non-canonical pathway	Caspase-11 activation		LPS binding to caspase-11 - Bacterial intracellular LPS exposure through GBPs and IRGB10 cooperation in type I IFN dependant-manner	LPS binding to caspase-11 - Caspase-11 activation dependent to type I IFN signalling during fungal infection
	GSDMD cleavage		Caspase-11-dependent activation	Caspase-11-dependent activation
	NLRP3 activation		K^+^ efflux through GSDMD pores	K^+^ efflux through GSDMD pores
	NLRP3-dependent cytokine secretion (on a per cell basis)		+++	+
	Cell death		GSDMD-dependent pyroptosis	Extrusion of chromatin through NETosis process with GSDMD-dependent mechanism

+ and +++: The amount of inflammasome-dependent cytokines released. Background color: It highlights the titles of the columns.

## Data Availability

Not applicable.

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
