# Peer review of "Specific NLRP3 Inflammasome Assembling and Regulation in Neutrophils: Relevance in Inflammatory and Infectious Diseases"

_cells, 2022, doi:10.3390/cells11071188_

Round 1
Reviewer 1 Report
The authors summarize evidence for the unusual assembly of NLRP3 inflammasome in neutrophils. They list prior studies on the inflammasome and its assembly, neutrophils, regulation of NLRP3 in neutrophils (canonical, non-canonical, and alternative pathways), inflammasome-mediated cell death in neutrophils, relevance of NLRP3 inflammasome activation in inflammation and infections. The authors provided three interesting figures that summarize NLRP3 inflammasome assembly, mechanisms of NLRP3 inflammasome activation in neutrophils, and the mechanisms of GSDMD-mediated NETosis. From this discussion, the authors suggest that an unusual assembly of the NLRP3 inflammasome takes place in neutrophils in response to various stimuli. Moreover, the aberrant activation of NLRP3 inflammasome is implicated in the pathogenesis of inflammatory diseases and infections. While the summarized information is interesting and the review is well-written, the authors do not provide take-home messages at the end of the respective section for the reader.
Comments:
1) In order to attract the interest of more readers regarding the clinical relevance of the work, the title of the present review needs to be modified to describe its relevance in inflammation and infectious diseases. I suggest being modified to:
“Unusual NLRP3 inflammasome assembling and regulation in neutrophils: Relevance in inflammatory and infectious diseases”.
2) In Figure 1:
- Authors are advised to make the figure caption stand-alone. To this end, authors are advised to provide the full names of all the listed abbreviations in the figures. In Figure 1, the full name of GSDMD needs to be described.
- To make Figure 1 more informative, authors are advised to add a visual distinction describing the title of each section on the top of each part of figure 1. For example, A. inflammasome structure; B., canonical pathway; C., non-canonical pathway.
3) In Figure 2:
- The full name of P2X7R and GSDMD need to be described.
- To make Figure 2 more informative, authors are advised to add a visual distinction describing the title of each section on the top of each part of figure 2. For example, A. canonical pathway; B. non-canonical pathway; C. resistance to pyroptosis.
4) In Figure 3:
The full name of GSDMD, GSDMDNT, and NETosis need to be described.
- To make Figure 2 more informative, authors are advised to add a visual distinction describing the title of figure 3; Mechanism of GSDMD-mediated NETosis.
5) The provided sections read like narration for the evidence of discussed points without critical aspects/reflection points. At the end of each section, a take-home message is advised to be provided.
6) More recent 2022 references are advised to be added to the review.
Author Response
We are very grateful for the work performed by reviewer#1 to improve our review article and for stating that our review “is interesting and the review is well-written”. Attached below are our responses to the suggestions and issues raised by reviewer#1.
Reviewer 1:
The authors summarize evidence for the unusual assembly of NLRP3 inflammasome in neutrophils. They list prior studies on the inflammasome and its assembly, neutrophils, regulation of NLRP3 in neutrophils (canonical, non-canonical, and alternative pathways), inflammasome-mediated cell death in neutrophils, relevance of NLRP3 inflammasome activation in inflammation and infections. The authors provided three interesting figures that summarize NLRP3 inflammasome assembly, mechanisms of NLRP3 inflammasome activation in neutrophils, and the mechanisms of GSDMD-mediated NETosis. From this discussion, the authors suggest that an unusual assembly of the NLRP3 inflammasome takes place in neutrophils in response to various stimuli. Moreover, the aberrant activation of NLRP3 inflammasome is implicated in the pathogenesis of inflammatory diseases and infections. While the summarized information is interesting and the review is well-written, the authors do not provide take-home messages at the end of the respective section for the reader.
Comments:
1) In order to attract the interest of more readers regarding the clinical relevance of the work, the title of the present review needs to be modified to describe its relevance in inflammation and infectious diseases. I suggest being modified to:
“Unusual NLRP3 inflammasome assembling and regulation in neutrophils: Relevance in inflammatory and infectious diseases”.
We thank reviewer #1 for this suggestion to improve the title of our review to include its relevance in diseases. We also have taken into account reviewer#2’s suggestion to propose the following title for our revised manuscript:
“Specific NLRP3 inflammasome assembling and regulation in neutrophils: Relevance in inflammatory and infectious diseases”
2) In Figure 1:
- Authors are advised to make the figure caption stand-alone. To this end, authors are advised to provide the full names of all the listed abbreviations in the figures. In Figure 1, the full name of GSDMD needs to be described.
- To make Figure 1 more informative, authors are advised to add a visual distinction describing the title of each section on the top of each part of figure 1. For example, A. inflammasome structure; B., canonical pathway; C., non-canonical pathway.
We agree with reviewer #1 and we have amended the figure and related legend accordingly.
3) In Figure 2:
- The full name of P2X7R and GSDMD need to be described.
- To make Figure 2 more informative, authors are advised to add a visual distinction describing the title of each section on the top of each part of figure 2. For example, A. canonical pathway; B. non-canonical pathway; C. resistance to pyroptosis.
We thank reviewer#1 for pointing this out. The figure 2 and its related legend have been amended accordingly.
4) In Figure 3:
The full name of GSDMD, GSDMDNT, and NETosis need to be described.
- To make Figure 2 more informative, authors are advised to add a visual distinction describing the title of figure 3; Mechanism of GSDMD-mediated NETosis.
We thank the reviewer for pointing this out. Figure legend 3 has been updated accordingly. However, we preferred not to implement the title in the body of figure 3 to avoid any redundancy with the title of the figure legend.
5) The provided sections read like narration for the evidence of discussed points without critical aspects/reflection points. At the end of each section, a take-home message is advised to be provided.
We thank reviewer#1 for this suggestion that will help readers. Thus, we have updated the end of sections accordingly.
6) More recent 2022 references are advised to be added to the review.
We thank the reviewer#1 for pointing this important omission. We have followed the suggestion from reviewer#1 and discussed the recent reference from Aymonnier et al., [1] highlighting ASC speck formation in the neutrophils of severe COVID-19 patients and correlation with NETosis.
Aymonnier, K., Ng, J., Fredenburgh, L.E., Zambrano-Vera, K., Münzer, P., Gutch, S., Fukui, S., Desjardins, M., Subramaniam, M., Baron, R., et al. (2022). Inflammasome activation in neutrophils of patients with severe COVID-19. Blood Advances bloodadvances.2021005949
Reviewer 2 Report
Comments:
This review manuscript focused on NLRP3 inflammasome in neutrophils, presented us a good summarization on the activation mechanisms as well as its associated diseases.
Suggestions:
1) For the first part: since NLRP3 inflammasome have been extensity reviewed in other published papers. There is no novelty in this section. So just a very brief introduction on NLRP3 activation will be fine.
2) The reviewer did not think NLPR3 activation in neutrophils is unusual based on the authors’ discussion. Maybe “specific” is better for the title.
3). The reviewer should present us a table side by side comparing NLRP3 inflammasome at least in macrophage and in neutrophils (expression levels, known signa 1 and 2, pyroptosis, and associated diseases). This will give readers more clear idea on the specific characterizations of NLRP3 inflammasome in neutrophils.
Author Response
We are very grateful for the work performed by reviewer#2 to improve our review article and for stating that our review represents “a good summarization on the activation mechanisms as well as its associated diseases”. Attached below are our responses to the suggestions and issues raised by reviewer#2.
Reviewer 2
This review manuscript focused on NLRP3 inflammasome in neutrophils, presented us a good summarization on the activation mechanisms as well as its associated diseases.
Suggestions:
1) For the first part: since NLRP3 inflammasome have been extensity reviewed in other published papers. There is no novelty in this section. So just a very brief introduction on NLRP3 activation will be fine.
We agree with reviewer #2’s suggestion to shorten the introduction on NLRP3 inflammasome. Instead, we have referred to comprehensive reviews on the topic (See below).
“The NLRP3 inflammasome has been actively reviewed, the readers should refer to more extensive reviews for more comprehensive details concerning the regulation of NLRP3 inflammasome activation [2–7].”
2) The reviewer did not think NLPR3 activation in neutrophils is unusual based on the authors’ discussion. Maybe “specific” is better for the title.
We thank reviewer #2 for pointing this out. We agree on have amended the title accordingly. We have also taken into account suggestion from the other reviewer to suggest the following title:
“Specific NLRP3 inflammasome assembling and regulation in neutrophils: Relevance in inflammatory and infectious diseases”
3). The reviewer should present us a table side by side comparing NLRP3 inflammasome at least in macrophage and in neutrophils (expression levels, known signa 1 and 2, pyroptosis, and associated diseases). This will give readers more clear idea on the specific characterizations of NLRP3 inflammasome in neutrophils.
We thank reviewer #2 for this great suggestion. We have now created a table (Table 1) that sum up the detailed mechanisms of activation of NLRP3 inflammasome in macrophages versus neutrophils. However, to render this comparison meaningful, we only have taken into account the - disease - models for which activation in both cell types have been investigated.